# Combining Artificial Neural Network and Driver–Pressure–State–Impact–Response Approach for Evaluating a Mediterranean Lake

Christos Tsitsis [1,*], Dimitrios E. Alexakis [2], Konstantinos Moustris [1] and Dimitra E. Gamvroula [2]

1 Laboratory of Air Pollution, Department of Mechanical Engineering, University of West Attica, 250 Thivon & P. Ralli Street, 12241 Athens, Greece
2 Laboratory of Geoenvironmental Science and Environmental Quality Assurance, Department of Civil Engineering, School of Engineering, University of West Attica, 250 Thivon & P. Ralli Street, 12241 Athens, Greece
* Correspondence: c.tsitsis@uniwa.gr

**Abstract:** The main objective of this research was to evaluate the surface water system of Lake Vegoritida (Region of Central Macedonia, Greece). The Driver–Pressure–State–Impact–Response (DPSIR) methodological approach was used. The analysis includes data from three (3) stations monitoring point source pollution and recording the most critical water quality measurement parameters in a time series data analysis from 1983 to 1997. The data will contribute to the analysis and was used to investigate, identify, and evaluate possible sources of chemical and ecological changes recorded in the lake. The artificial neural network (ANN) is a valuable tool for making predictions based on the water quality data set. The findings highlighted the increased concentration of nutrients that contribute to the presence of eutrophic conditions, while their seasonal variability is mainly due to factors, such as water level fluctuations and biological processes in the lake. The above, combined with the critical biotic indicators and factors alongside the reduction in biodiversity, indicated that only the most resistant species survive, confirming the previous finding. In Greece, systematic monitoring and reporting programs have recently been implemented, such as the ECOFRAME scheme and the guidelines proposed by the "Intercalibration Group for Mediterranean Lakes". The water quality status could be classified as "High", "High to Good", and "High to Poor", respectively, while the overall ecological assessment tends to change to poor conditions. The actions required at an early stage concern the planning of programs and actions that contribute to the sustainable management of land uses and the reduction in point sources of pollution, as well as the reduction of the applied quantities of agrochemicals on the cultivated land in the study area.

**Keywords:** WFD; ECOFRAME system; multi-layer perceptron artificial neural networks; surface water; Lake Vegoritida; Greece





## 1. Introduction

Critical water resource problems are emerging in the Mediterranean region due to excess nutrients and pesticide leaching, which cause adverse environmental degradation and eutrophication in freshwater ecosystems and seawater inflow into aquifer systems, causing deterioration of the water quality [1–9]. Since the presence of chemical compounds in water significantly impacts its use for human consumption and agriculture, the evaluation of water quality applying complicated methods and approaches has become an important topic worldwide [10–13]. This study aims to evaluate the ecological status of Lake Vegoritida by applying the Driving Forces–Pressures–State–Impact–Response (DPSIR) approach [14,15] and artificial neural network (ANN) as a predictor, prevention and troubleshooting tool in focal stations for measuring and receiving data. The findings are expected to lead to adopting measures and policies that will contribute to evaluating the sustainable management of water bodies. Communities and their activities rapidly increase

the pressures exerted on the water bodies. Therefore, it is essential to adopt practices to maintain the quality of the ecological elements of the region to ensure sustainability, the quality of water resources, and the protection of the natural environment, as well as a more balanced coexistence of the natural component in combination with human activities.

The typification of the ecological condition of lakes has become a mandatory legal guideline after the adoption of the water framework directive (WFD) by the European Member States [16]. WFD was established to create the mechanisms required to maintain and manage groundwater, fresh water and marine water in the European Union in a sustainable state by 2015. Furthermore, WFD also legislates a comprehensive framework for water management, including measures to prevent and maintain the quality of water bodies, the rational usage of water, and the increased protection and improvement of the aquatic environment. Many surface fresh waters bodies worldwide face dramatic environmental degradation with downward trends in water quality degradation. Various studies predict that many lakes in the European region will encounter difficulties in implementing the required WFD criteria by 2015. Therefore, it is important to adopt the best approaches and procedures for restoration and management [17]. Anagnostidis and Economou [18] and Benedini and Tsakiris [19] pointed out the difference between the Mediterranean and other temperate lentic systems, considering lake systems, studying the climate variability, and considering the climatic specificity of the Mediterranean climate [16,20,21]. In addition, many studies measuring trace elements show the degradation of water quality on a global scale due to various natural and socioeconomic sources [22,23]. The European Water Policy, especially the WFD, implemented methodological approach tools that contribute to the sustainable management of aquatic resources, such as the DPSIR methodology, which was established as a standardized framework for identifying pressures and effects in the context of WFD [24,25]. According to DPSIR, the approach involves a sequence of linkages that include "drivers" (reason) through "pressures" (e.g., pollutants) to "states" (physical, chemical, biological) and "effects" on ecosystems (structure and operation) and finally drives to "responses" (policy) [14,17,26,27]. It was originally developed as a methodology framework dealing with environmental problems (Stress–Response) and then adopted by the Organization for Economic and Cooperative Development (OECD) as Pressures–State–Response (PSR) [28]. The final DPSIR structure has been adopted by the European Environment Agency as an auxiliary management tool through the presentation of indicators, additionally providing the possibility of feedback in policy making on issues of the protection and preservation of environment quality [17]. The investigated water body (Lake Vegoritida) has a chronic history of eutrophication due to diverse anthropogenic pressures [29]. Although improvement efforts have been made, such as reducing the external burden from anthropogenic activities, the lake has an ecosystem of poor biological quality; at the same time, many activities (irrigation, fishing, tourism, and recreation) are still permitted [30]. The possibility that the ongoing activities involved with the studied lake will reduce or increase the recovery process is an open question of investigation that clearly needs the involvement to involve the formulation of contemporary management policies in conjunction with comprehensive programs to assess the pressures on the water body. The aim of this research is to apply a supervisory control and forecasting model using ANN to estimate the water quality condition of a surface water body.

## 2. Methodology

### 2.1. Study Region and Data Compilation

Lake Vegoritida is located in NW Greece (Figure 1). It is a shallow Mediterranean lake with an average depth of about 0.4 m and a maximum depth of 26.6 m. The lake has a surface area of approximately 40 km$^2$ and is situated in the broader area of Ptolemaida and Amuntaio cities (15,000 inhabitants); approximately 60% of the primary use in the lake's catchment is mainly used for agricultural operations. The watershed has no physical outflows and is recharged by karst sources. The lake has a long history of eutrophication due to intense point and non-point nutrient loading occurring since 1978 [29]. The rehabilitation

applications reduced the body's external loads to a significant extent, but it still maintained high P-loading to maintain eutrophic conditions [29]. According to Habitats Directive 92/43/EEC [31] for preserving the ecological status, Lake Vegoritida is included in the network of the Special Conservation Areas Natura 2000. However, the characterization of the ecological status of the lake has become a legal obligation after the introduction of the WFD [31].

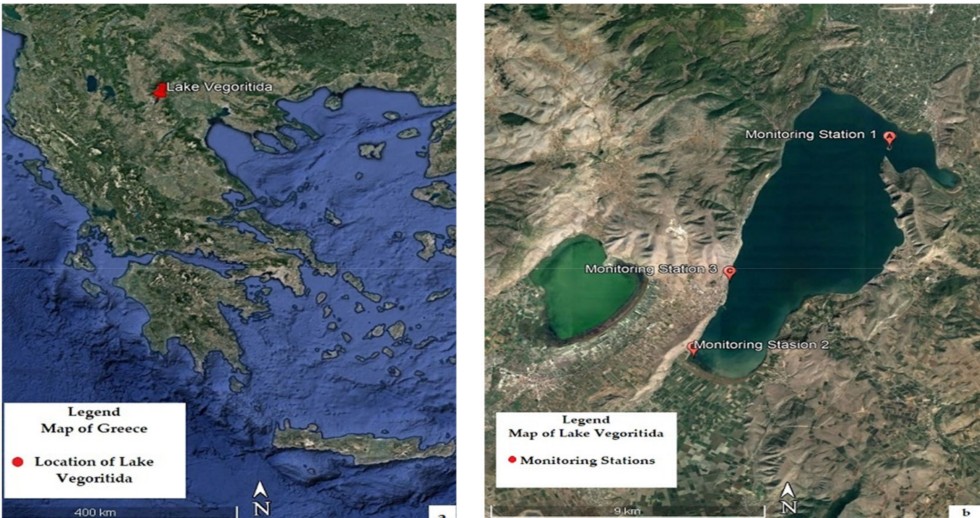

**Figure 1.** (**a**) Map of Greece presenting the location of Lake Vegoritida; (**b**) Map of study area showing the location of monitoring stations (modified from Google Earth [32]).

The data were from relevant databases, bibliographic sources, and investigation projects, including national monitoring programs from the Greek Ministry of Agriculture [30,33–35]. Monthly time-step data of the following water quality physicochemical parameters were available from 1983 to 1997 with monthly time-step. The following water quality parameters: total nitrogen (TN) mg $L^{-1}$, total phosphorous (TP) mg $L^{-1}$, dissolved oxygen (DO) mg $L^{-1}$, pH, and electrical conductivity (CND) $\mu$S cm$^{-1}$. Many gaps in the data series of the chemical and physical parameters were noticed. The methodology of an artificial neural network was selected for the prediction of TP concentration in lake water. The specific biochemical index is essential for the assessment of the water body.

*2.2. Evaluation of a Water Body*

In the last 30 years, water quality evaluation and measurement techniques have been using indicators (bioindicators), since the plant, invertebrate, and fish species interact and affect water quality in ecosystems. In addition, it is also a tool for investigating the changes in an ecosystem's environmental quality [36]. Collecting and identifying indicator species belonging to different biological groups (algae, macrophytes, inland water invertebrates, and fish) provides an opportunity to assess long-term water and ecosystem quality alteration [37]. In Greece, the incomplete environmental data and the limited use of monitoring programs, mainly concerning the biological data proposed by the WFD, create significant difficulties in characterizing the ecological conditions of water bodies [35]. Therefore, the choice of indicators for the characterization of the lake is based on: (a) the requirements of the WFD; (b) the interaction with the phenomenon of eutrophication; and (c) the available data that can be obtained [5]. In this study, the characterization of Lake Vegoritida followed the typological criteria given by the WFD, as well as the ECOFRAME system [6,15,29,37]. It is important to emphasize that we find lakes with completely different geographical characteristics in the European region.

The number and size of lakes within countries located in the European region varies greatly. In southern Europe, lakes are principally tank systems, mainly used for water storage and the supply water for agricultural purposes. In the southern Europe, these tanks additionally deal with the phenomenon of drought caused by climate change, whereas in the northernmost part of the continent, we find areas that are less populated, with lower pressure on the water bodies, combined with the use of environmentally friendly conservation plans, which leads to the preservation of the lakes at a high level [38]. In other countries, mainly on the Asian continent, lakes show a long history of pollution and/or eutrophication with damaged habitats [39–41]. Greek lakes are ranked in the second group of European lakes. The climate, the depletion of water resources from exploitation and use, and the summer drought exert significant pressure on the hydrology of fresh waters, specifically lakes [42]. As stated by Premazzi and Chiaudani [43], significant problems that have widely affected the lakes and their ecological condition are the following: (a) the phenomenon of eutrophication as a result of an excessive input of nutrients and organic matter; (b) the fluctuation of the water level as a result of natural and anthropogenic factors; (c) insufficient watershed erosion control; and (d) acidification and contamination by toxic substances. In Greece, in recent decades, increased urbanization, wastewater disposal, wetland regulation, and more intensive agricultural practices have increased nutrient loading in many lakes. Furthermore, Greek lakes face a continuous decrease in water levels [42], which can lead to state changes and affect the functioning of biotic components [44]. The inflow of suspended particulate matter from agricultural lands and the widespread presence of toxic substances from activities in freshwater bodies are also common practices [45].

### 2.3. Artificial Neural Networks

Early in the 1950s, the development of artificial neural networks (ANNs) as a branch of artificial intelligence begins. It was the first step in mimicking the human brain's biological structure through mathematical functions. Typical ANNs use elementary neuron models. Artificial neuron models retain only very coarse features of biological neurons [46]. The scientific interest in the operation and applications of computational systems development, along with the rapid analysis and processing of information in recent decades, has increased the use of ANNs in many applications and issues. ANN has been developed exponentially during the last decades, primarily due to the availability of suitable computing systems that result in fast data analysis and information processing [47]. This study employed ANN, known as multi-layer-perceptron artificial neural networks (MLP-ANN). MLP-ANN includes one input layer, one or more invisible layers, and the output layer where the results targets are extracted. The input data through the input layer passes its values to the first hidden layer where appropriate processing is performed. Their results are sent to the second hidden layer, where they are further processed. This operation is followed for all successive hidden layers until the generated result is passed to the result extraction layer (Figure 2).

In this particular work, an ANN was developed and trained to predict the average annual value of TP concentration (mg L$^{-1}$) through the wet season of the year (October–April) in Lake Vegoritida, Greece. The developed ANN belongs to the category of MLP-ANN. The data used to train the specific ANN were the mean annual values of the studied parameters (T, TN, TP, DO, pH, and CND). All the values mentioned earlier were obtained from sampling three different places of Lake Vegoritida from 1983–1997. After the training of the ANN, the latter was capable of providing a complete time series of average annual total phosphorus concentration values for Lake Vegoritida for the entire period between 1983–1997.

The final architecture of the developed ANN model consists of an input layer with three (3) input artificial neurons, a hidden layer with three (3) hidden ANNs, and an output layer with one ANN corresponding to the rate of mean annual TP concentration in water of Lake Vegoritida. The selection of the appropriate input data and the architecture of the

developed ANN was found after iterative application of the trial and error method [48,49]. The value of the determination coefficient ($R^2$) is equal to 0.962, which shows that the developed ANN model can interpret 96.2% of the variation in the average annual TP concentration in the water of Lake Vegoritida (Figure 3). The agreement index (IA) value is equal to 0.986 and very close to unity, indicating that the predicted values of the average annual TP concentration in water from the model are very close to the corresponding observed values.

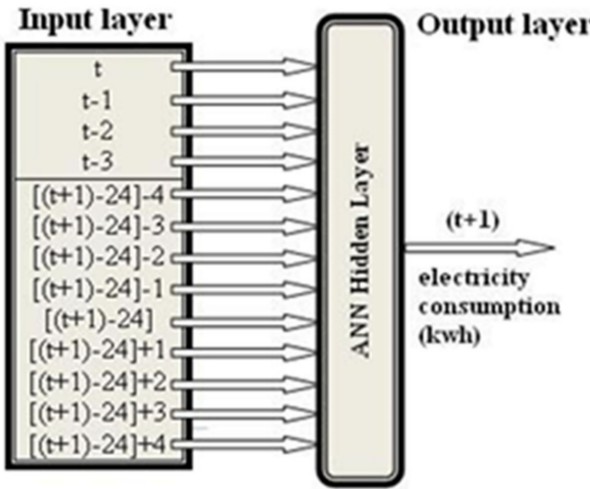

**Figure 2.** Typical architecture of a feed-forward multi-layer perceptron artificial neural (MLP-ANN) (modified from [47]).

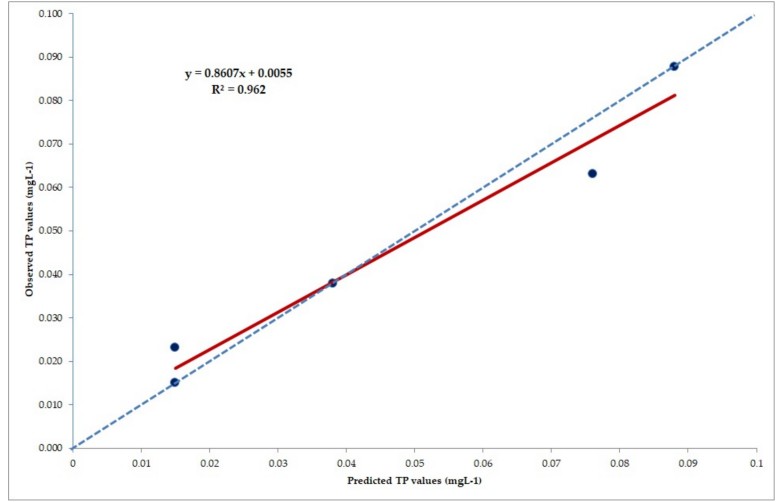

**Figure 3.** Determination coefficient between observed and predicted TP values by ANN.

## 3. Results and Discussion

### 3.1. Physico-Chemical Analysis

Two classification systems were proposed for evaluating the water quality data of Lake Vegoritida, the Mediterranean GIG classification [21] and ECOFRAME [27]. For the more rational use and data analysis, two smaller sets were created: (a) wet season; and (b) dry season. Based on the climatic conditions of the area studied, the hydrological year includes a wet and dry season. The wet season runs from October to March, while the dry season is from April to September. Each season's average values were calculated from monthly monitoring (Figure 4, Table 1). For average temperature values, the highest average water temperature (24.2 °C) was recorded in the dry season of 1995.

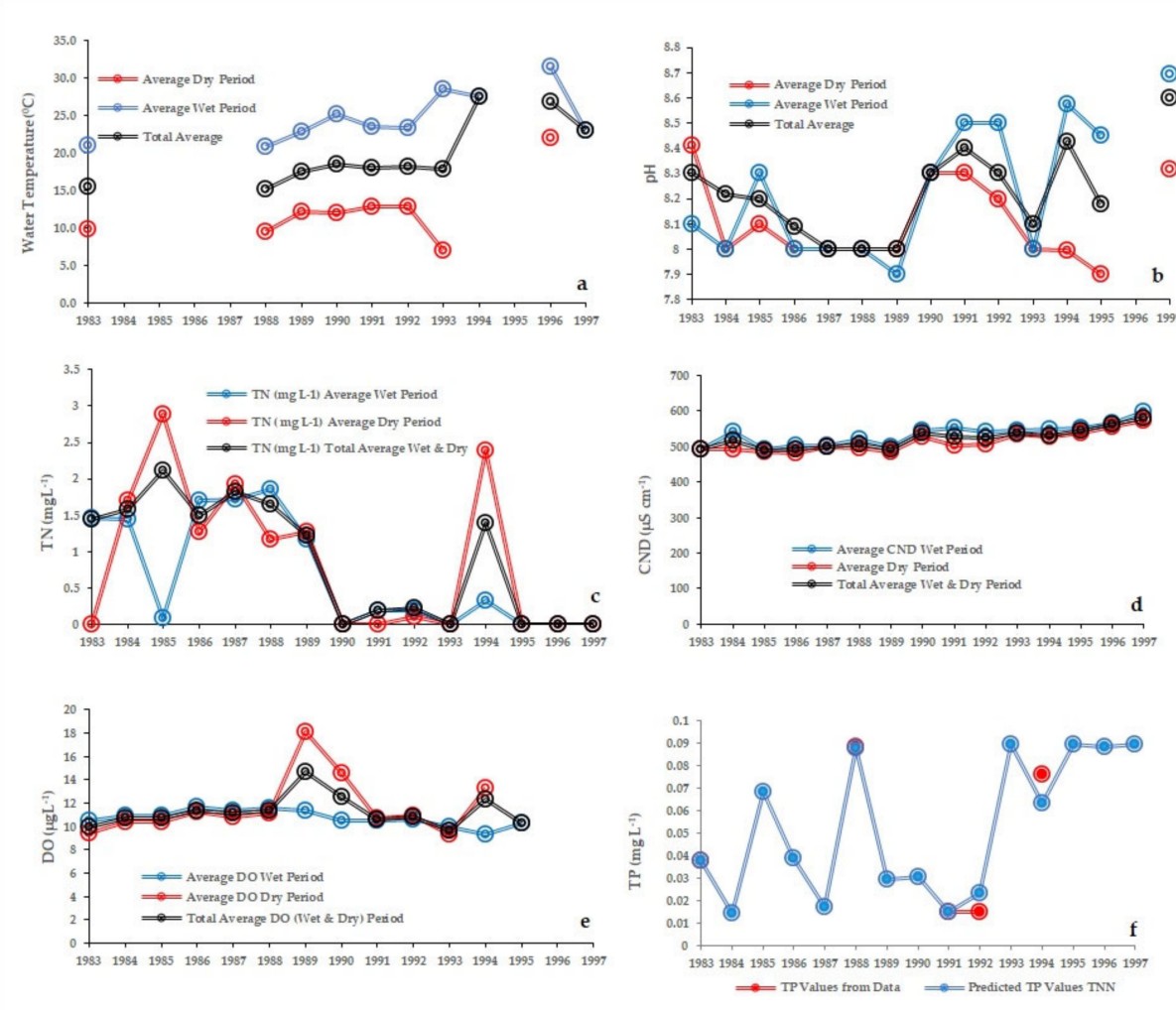

**Figure 4.** Time series of selected water quality parameters of Lake Vegoritida during the monitoring period 1983–1997: (**a**) water temperature; (**b**) pH; (**c**) CND; (**d**) DO; (**e**) TN; and (**f**) TP (data were provided by HMRDF 2019).

**Table 1.** Lowest and highest yearly average values of the studied water quality dataset.

| | T | pH | CND | DO | TN | TP |
|---|---|---|---|---|---|---|
| | (°C) | | ($\mu$S cm$^{-1}$) | (mg L$^{-1}$) | (mg L$^{-1}$) | (mg L$^{-1}$) |
| Lowest | 8.0 | 7.7 | 484 | 9.0 | 0.077 | 14 |
| Highest | 24.2 | 8.5 | 597 | 18.0 | 1.84 | 90 |
| Number of yearly average values | 13 | 14 | 14 | 14 | 14 | 10 |

In comparison, the lowest average water temperature (8.0 °C) was observed in the wet season of 1994 (Figure 4). The highest pH average value (8.5) was during the season of 1995 and the lowest pH average value (7.9) was during the period of 1993 (Figure 4). The maximum DO average concentration (18.0 mg L$^{-1}$) was observed through the dry season of 1989, while the minimum average concentration (9.0 mg L$^{-1}$) was recorded through the dry season of 1993 (Figure 4). All the average DO concentrations observed during the dry season are lower than those recorded during the wet season, which happens in cases where high temperatures were recorded, especially during summer months (up to 24.6 °C). This mainly changes the DO concentration. The concentration of DO is a parameter that

contributes to preserving the life of the organisms within Lake Vegoritida, according to Chang [3] and Novotny [22]. Concerning conductivity values, a maximum average value (597 µS cm$^{-1}$) was observed through the wet season of 1997, and the minimum average value (484 µS cm$^{-1}$) was recorded in the dry season of 1989, while there is no evident change in the CND average values through the wet and dry seasons of the examined years (Figure 4). The minimum TN average concentration (0.077 mg L$^{-1}$) in the water of Lake Vegoritida was recorded through the dry season of 1985, while the minimum TP average concentration (15 mg L$^{-1}$) in the water of Lake Vegoritida was recorded through the dry season of 1991. In addition, the maximum TN average concentration (1.84 mg L$^{-1}$) was observed through the dry season of 1988, and the maximum TP average concentration (90 mg L$^{-1}$) in the water of Lake Vegoritida was recorded through the wet season of 1993. The low water levels that were observed during dry seasons recorded high TN and TP loads were mainly produced by agricultural activities in combination with high water temperature, generating plant biomass production and increasing the rates of the denitrification process in the water body [14]. Increased average TP content in the water of Lake Vegoritida were mainly recorded through the dry seasons of the year (Figure 4). Moss et al. [2] reported that increasing TP concentration in shallow lake water during summer might be an additional effect of phosphate release.

### 3.2. Correlation Analysis

Statistical analysis was performed to test the correlation between parameters for the wet and dry periods. Pearson correlation coefficient (r) is calculated by applying the software platform Microsoft® Excel. The prediction technique with ANN is used for the missing values of total phosphorus (TP). All statistical analyses were performed with Microsoft Excel® (Redmond, Washington, DC, USA) and IBM SPSS® Statistics (International Business Machines Corporation, Statistical Product and Service Solutions, Armonk, NY, USA).

The statistical analysis for the wet period between 1983–1997 suggests that the indicators of pH and TP present a good correlation with each other (r > 0.68; $p < 0.05$) (Table 2). In addition, the CND, Na, DO, and NO$_3^-$ group show moderate correlation (r > 0.65; $p < 0.05$). The SO$_4^{2-}$ and Mg group present a moderate correlation (r > 0.66; $p < 0.05$). The group of N-NO$_3$, CND, NO$_3^-$, and N-NH$_4$, presents a high correlation (r > 0.78; $p < 0.05$) (Table 2).

The statistical analysis for the dry period between 1983–1997 suggests that the indicators of pH, CND, Na, and N–NO$_3^-$ present a strong correlation (r > 0.79; $p < 0.05$) (Table 3). The Cl and Mg indicator group shows a good correlation (r > 0.74; $p < 0.05$). The indicator group of Na, CND, and N–NO$_3$ presents a good correlation (r > 0.65; $p < 0.05$). Finally, the indicator group of pH and N-NO$_3$ delivers a high correlation (r > 0.89; $p < 0.05$).

### 3.3. Typification of Lake Vegoritida

Reference conditions are defined based on any deviation in the ecological status and water quality related to the change in conditions that vary according to the type of lake (shallow, deep, cold, hot, small, large, and high- and low-altitude lakes) [24]. Greece's characteristics and reference conditions have been established recently [50]. As reported by the ECOFRAME system, Lake Vegoritida was classified as a "warm" lake as the season with low temperatures that creates ice cover is less than two months a year and the average temperature of the hottest month is greater than 25 °C [9,35,51]. The typology plan followed the WFD Appendix II typology criteria, which includes dimensions, altitude, geology, and deepness (System A), as well as selected System B criteria and the other additional criteria (Table 4) [52]. Appendix II of the WFD proposes two options to define lake typology. The first option is System A, based upon ecoregions (Appendix XI) and the obligatory factors, such as geology, size, altitude, and depth. System B is the second option, which includes a wide range of optional factors (Appendix II) [52].

**Table 2.** Correlation coefficients and *p*-values of the studied water quality parameters for the wet period 1983–1997.

| | | pH | CND | Cl | SO$_4$ | Na | Mg | DO | N–NO$_3$ | N–NO$_2$ | N–NH$_4$ | TP |
|---|---|---|---|---|---|---|---|---|---|---|---|---|
| pH | r | 1 | 0.127 | −0.365 | 0.369 | 0.091 | 0.282 | −0.296 | −0.544 | −0.612 | −0.768 | −0.683 |
| | *p*-value | | 0.665 | 0.199 | 0.194 | 0.757 | 0.329 | 0.326 | 0.13 | 0.144 | 0.075 | 0.007 |
| | N | 14 | 14 | 14 | 14 | 14 | 14 | 13 | 9 | 7 | 6 | 14 |
| CND | r | 0.127 | 1 | 0.392 | 0.507 | 0.642 | 0.334 | −0.592 | −0.716 | 0.214 | −0.613 | 0.384 |
| | *p*-value | 0.665 | - | 0.148 | 0.054 | 0.01 | 0.224 | 0.033 | 0.03 | 0.646 | 0.196 | 0.157 |
| | N | 14 | 15 | 15 | 15 | 15 | 15 | 13 | 9 | 7 | 6 | 15 |
| Cl$^-$ | r | −0.365 | 0.392 | 1 | −0.411 | 0.098 | −0.13 | 0.01 | 0.343 | 0.114 | 0.509 | 0.434 |
| | *p*-value | 0.199 | 0.148 | - | 0.128 | 0.728 | 0.656 | 0.975 | 0.366 | 0.809 | 0.303 | 0.106 |
| | N | 14 | 15 | 15 | 15 | 15 | 15 | 13 | 9 | 7 | 6 | 15 |
| SO$_4$$^{2-}$ | r | 0.369 | 0.507 | −0.411 | 1 | 0.364 | 0.66 | −0.133 | −0.646 | 0.048 | −0.666 | −0.182 |
| | *p*-value | 0.194 | 0.054 | 0.128 | - | 0.183 | 0.008 | 0.665 | 0.06 | 0.919 | 0.148 | 0.517 |
| | N | 14 | 15 | 15 | 15 | 15 | 15 | 13 | 9 | 7 | 6 | 15 |
| Na$^+$ | r | 0.091 | 0.642 | 0.098 | 0.364 | 1 | −0.12 | −0.198 | −0.674 | 0.147 | −0.493 | 0.365 |
| | *p*-value | 0.757 | 0.01 | 0.728 | 0.183 | - | 0.654 | 0.516 | 0.046 | 0.753 | 0.321 | 0.18 |
| | N | 14 | 15 | 15 | 15 | 15 | 15 | 13 | 9 | 7 | 6 | 15 |
| Mg$^{2+}$ | r | 0.282 | 0.334 | −0.125 | 0.657 | −0.126 | 1 | −0.171 | −0.56 | 0.242 | −0.615 | −0.219 |
| | *p*-value | 0.329 | 0.224 | 0.656 | 0.008 | 0.654 | - | 0.577 | 0.117 | 0.601 | 0.194 | 0.434 |
| | N | 14 | 15 | 15 | 15 | 15 | 15 | 13 | 9 | 7 | 6 | 15 |
| DO | r | −0.296 | −0.592 | 0.01 | −0.133 | −0.198 | −0.17 | 1 | 0.765 | 0.113 | 0.578 | −0.18 |
| | *p*-value | 0.326 | 0.0331 | 0.975 | 0.665 | 0.516 | 0.577 | - | 0.016 | 0.809 | 0.229 | 0.555 |
| | N | 13 | 13 | 13 | 13 | 13 | 13 | 13 | 9 | 7 | 6 | 15 |
| N-NO$_3$ | r | −0.544 | −0.716 | 0.343 | −0.646 | −0.674 | −0.56 | 0.765 | 1 | −0.064 | 0.974 | 0.164 |
| | *p*-value | 0.13 | 0.03 | 0.366 | 0.06 | 0.046 | 0.117 | 0.016 | - | 0.892 | 0.005 | 0.673 |
| | N | 9 | 9 | 9 | 9 | 9 | 9 | 9 | 9 | 7 | 5 | 9 |
| N-NO$_2$ | r | −0.612 | 0.214 | 0.114 | 0.048 | 0.147 | 0.242 | 0.113 | −0.064 | 1 | −0.563 | 0.308 |
| | *p*-value | 0.144 | 0.646 | 0.809 | 0.919 | 0.753 | 0.601 | 0.809 | 0.892 | - | 0.619 | 0.502 |
| | N | 7 | 7 | 7 | 7 | 7 | 7 | 7 | 7 | 7 | 3 | 7 |
| N-NH$_4$ | r | −0.768 | −0.613 | 0.509 | −0.666 | −0.493 | −0.61 | 0.578 | 0.974 | −0.563 | 1 | 0.395 |
| | *p*-value | 0.075 | 0.196 | 0.303 | 0.148 | 0.321 | 0.194 | 0.229 | 0.005 | 0.619 | - | 0.438 |
| | N | 6 | 6 | 6 | 6 | 6 | 6 | 6 | 5 | 3 | 6 | 6 |
| TP | r | −0.683 | 0.384 | 0.434 | −0.182 | 0.365 | −0.22 | −0.18 | 0.164 | 0.308 | 0.395 | 1 |
| | *p*-value | 0.0071 | 0.157 | 0.106 | 0.517 | 0.18 | 0.434 | 0.555 | 0.673 | 0.502 | 0.438 | - |
| | N | 15 | 15 | 15 | 15 | 15 | 15 | 13 | 9 | 7 | 6 | 15 |

**Table 3.** Correlation coefficients and *p*-values of the studied water quality parameters for the dry period 1983–1997.

| | | pH | CND | Cl | SO$_4$ | Na | Mg | DO | N–NO$_3$ | N–NO$_2$ | N–NH$_4$ |
|---|---|---|---|---|---|---|---|---|---|---|---|
| pH | r | 1 | 0.714 | −0.117 | 0.446 | 0.760 | −0.03 | −0.107 | 0.892 | 0.022 | −0.253 |
| | *p*-value | - | 0.004 | 0.69 | 0.127 | 0.003 | 0.923 | 0.74 | 0.007 | 0.967 | 0.837 |
| | N | 14 | 14 | 14 | 13 | 13 | 13 | 12 | 7 | 6 | 3 |
| CND | r | 0.714 | 1 | 0.454 | 0.540 | 0.666 | 0.43 | −0.373 | 0.513 | 0.394 | −0.94 |
| | *p*-value | 0.004 | - | 0.089 | 0.046 | 0.009 | 0.125 | 0.233 | 0.239 | 0.44 | 0.221 |
| | N | 14 | 15 | 15 | 14 | 14 | 14 | 12 | 7 | 6 | 3 |
| Cl$^-$ | r | −0.117 | 0.454 | 1 | −0.109 | 0.11 | 0.744 | −0.244 | −0.309 | −0.434 | - |
| | *p*-value | 0.69 | 0.089 | - | 0.711 | 0.709 | 0.002 | 0.444 | 0.5 | 0.39 | 0 |
| | N | 14 | 15 | 15 | 14 | 14 | 14 | 12 | 7 | 6 | 3 |
| SO$_4$$^{2-}$ | r | 0.446 | 0.540 | −0.109 | 1 | 0.647 | 0.239 | 0.057 | −0.287 | 0.1 | −0.978 |
| | *p*-value | 0.127 | 0.046 | 0.711 | - | 0.012 | 0.411 | 0.868 | 0.532 | 0.85 | 0.135 |
| | N | 13 | 14 | 14 | 14 | 14 | 14 | 11 | 7 | 6 | 3 |
| Na$^+$ | r | 0.760 | 0.666 | 0.11 | 0.647 | 1 | 0.431 | 0.372 | −0.014 | 0.27 | −0.974 |
| | *p*-value | 0.003 | 0.009 | 0.709 | 0.012 | - | 0.124 | 0.26 | 0.976 | 0.605 | 0.146 |
| | N | 13 | 14 | 14 | 14 | 14 | 14 | 11 | 7 | 6 | 3 |
| Mg$^{2+}$ | r | −0.03 | 0.43 | 0.744 | 0.239 | 0.431 | 1 | 0.229 | −0.539 | −0.692 | −0.93 |
| | *p*-value | 0.923 | 0.125 | 0.002 | 0.411 | 0.124 | | 0.498 | 0.212 | 0.128 | 0.239 |
| | N | 13 | 14 | 14 | 14 | 14 | 14 | 11 | 7 | 6 | 3 |
| DO | r | −0.107 | −0.373 | 0.244 | 0.057 | 0.372 | 0.229 | 1 | −0.174 | −0.257 | −0.886 |
| | *p*-value | 0.74 | 0.233 | 0.444 | 0.868 | 0.26 | 0.498 | - | 0.709 | 0.623 | 0.306 |
| | N | 12 | 12 | 12 | 11 | 11 | 11 | 12 | 7 | 6 | 3 |
| N-NO$_3$ | r | 0.892 | 0.513 | −0.309 | −0.287 | −0.014 | −0.539 | −0.174 | 1 | 0.039 | 1.000 |
| | *p*-value | 0.007 | 0.239 | 0.5 | 0.532 | 0.976 | 0.212 | 0.709 | - | 0.941 | - |
| | N | 7 | 7 | 7 | 7 | 7 | 7 | 7 | 7 | 6 | 2 |
| N-NO$_2$ | r | 0.022 | 0.394 | −0.434 | 0.1 | 0.27 | −0.692 | −0.257 | 0.039 | 1 | −1.00 |
| | *p*-value | 0.967 | 0.44 | 0.39 | 0.85 | 0.605 | 0.128 | 0.623 | 0.941 | - | - |
| | N | 6 | 6 | 6 | 6 | 6 | 6 | 6 | 6 | 6 | 2 |
| N-NH$_4$ | r | −0.253 | −0.94 | - | −0.978 | −0.974 | −0.93 | −0.886 | 1.00 | −1.00 | 1 |
| | *p*-value | 0.837 | 0.221 | 0 | 0.135 | 0.146 | 0.239 | 0.306 | - | - | - |
| | N | 3 | 3 | 3 | 3 | 3 | 3 | 3 | 2 | 2 | 3 |

**Table 4.** Classification of Lake Vegoritida based on the WFD ECOFRAME scheme and comparative literature—MED-GIG system [29,33].

| | |
|---|---|
| Residential area | As stated by Appendix XI of the WFD, Lake Vegoritida belongs to the "Hellenic Western Balkan" residential area; as stated by System A of Appendix II of the WFD, three height categories are specified: plain (<200 m a.s.l), mid-height (200–800 m a.s.l.), and high height (>800 m a.s.l). Lake Vegoritida is classified as mid-height category, since it is localized at 515 m a.s.l. |
| Highest and average depth | System A of Appendix II of the WFD recommends three average depth categories, as follows: very shallow (<3 m), shallow (3–15 m) and deep lakes (15 m). Lake Vegoritida has a maximum depth ($Z_{max}$) of 26.0 m and an average depth ($Z_{average}$) of 15.0, which falls into to the shallow lake category; it also has ratio of $Z_{average}/Z_{max}$ that is equal to 0.57 |
| Surface area size | The lake has an area of 40 km$^2$ and it belongs to the "large class size", as stated by Appendix II while the watershed is about 1853 km$^2$ |
| Geology | The watershed of Lake Vegoritida shows corrosion in places where there is limestone. In addition, the sediments of the bottom of Lake Vegoritida include silt, sand, gravel, clay and sandy materials [53]. As stated by System A of Appendix II of the WFD, three types of geology are specified (calcareous, siliceous, and organic), and the Lake Vegoritida falls into the calcareous category |
| Mixing system | Lake Vegoritida is a polymixed lake that shows light stratifications during the summer months of the year [54] |

### 3.4. The DPSIR Analysis

The DPSIR factor methodology approach is described by indicators, which have two main objectives: (a) to reduce the number of parameters that change the factor chain system, and (b) to simplify the process from which information can be collected, and actions that contribute to the improvement of the systems arise [7]. For the study area, the DPSIR model presupposes the construction of a set of factors for each indicator, so that it is possible to analyze any socio-economic process that contributes to or affects the water body. In many cases, the evidence refers to pressures exerted on the system by the human factor, which changes the quality of lake ecosystems. In the first phase, the indicators selected mainly focused on water quality to fulfil each category of the DPSIR model under consideration. The environmental conditions in the whole watershed, as well as the ecological connection between environmental factors and living organisms. Considering the typological features which are presented in Table 5 and Figure 5 as well as the Mediterranean Lake Intercalibration Report, Lake Vegoritida is classified as a type B Mediterranean lake, which includes natural large, shallow, and mid-altitude lakes with a polymictic regime and human-controlled outlet [50].

### 3.5. Land-Use Changes, Agriculture, Livestock, and Urbanization

According to the data of the Hellenic Statistical Service [55] in the Municipality of Florina, Ptolemaida, and Kozani, which belongs to the basin of Lake Vegoritida, the primary crop is cereals, which occupy about 300,000 maize acres, followed by legumes (12,700), potatoes (5500), beets (9000), fodder plants (90,000), vines (10,500), and trees (21,000). A significant percentage of the population is engaged in animal husbandry, lignite mining, and electricity production [51]. The lignite mines in the Ptolemaida and Filota areas are also responsible for the significant drop in the lake level in previous years due to over-pumping. In the municipality of Amyntaio, the regions covered by agricultural activities occupy the most significant percentage (66%), followed by areas with forests and semi-natural areas (23%), areas covered by water (8%), and artificial regions (3%) [51]. In the Amyntaio and Ptolemaida area, 270 and 200 boreholes were constructed, respectively. Most of these boreholes are exploratory, while the exact number of productive boreholes is constantly changing.

**Table 5.** Dataset of the applied DPSIR indicators in the lake studied.

| Driver | Pressure | State | Impact | Response |
|---|---|---|---|---|
| Husbandry, Stockraising | -Fertilizer use, herbicide use, land use change, irrigation, stock raising, house urban waste, sewage treated effluents | - High TN content in lake water<br><br>- High annual fluctuation of the water level | - Habitat destruction<br><br>-Intense eutrophication<br><br>-Water toxicity | - Code of correct agricultural policies<br>- Implementation of WFD |
| Land use changes and urbanization | - Natural element<br><br>-Development of urban and sub-urban areas<br><br>- Pretension for wastewater treatment | -Increase pollutant load<br><br>-Degradation of water quality<br><br>-High content of dissolved oxygen | -Habitat loss<br><br>-Extinction of species<br><br>-Eutrophication | -Implementation of the WFD<br>-Evaluation of good operation sewage treatment plants<br>-Determination of residential development zones;<br>-Agricultural policies, management plans |
| Industrial development | -Industrial waste<br><br>Irrigation requirements | -Fluctuations of water layer<br>-Increasing trends of nitrates in water | -Water quality<br><br>- Habitat loss<br><br>-Conservation status | - Implementation of the WFD |
| Demand for protection of the ecosystem | -Demands for conservation, species conservation, climate change (floods/droughts) | Reduction in the wetland area<br><br>-Species populations are declining<br><br>-Indications of disappears for invasive species | -Biodiversity Effects | -Conservation measures for species and habitat<br><br>-Evaluation of goods and services offered by the ecosystem<br>-Participation of local municipalities in action plans |

### 3.6. Implications of the Ecosystem

The significant differences in water level significantly affect the habitats and social structure. Climate change has the potential to put further pressure on the lake's ecosystem while the spread of crops and urban areas has eliminated wetlands (e.g., wet meadows, hedgerows, pastures), with adverse effects on the entire ecosystem and especially on avifauna.

### 3.7. Assessment of the State and Impacts Analysis

Human intervention, practiced activities and various natural factors acting synergistically bring different pressures on ecosystems. Furthermore, ecosystems respond in a "context-dependent" manner to the conditions created by the above factors, making it difficult to forecast the effect of a given pressure accurately.

| Year | pH | TP[a] (mgL$^{-1}$) | TN[b] (mgL$^{-1}$) |
|---|---|---|---|
| 1983 | 8.1 | 38 | 1.46 |
| 1984 | 8.2 | 14 | 1.48 |
| 1985 | 8.1 | 68 | 0.077 |
| 1986 | 8 | 39 | 1.7 |
| 1987 | 8 | 17 | 1.72 |
| 1988 | 7.7 | 88 | 1.84 |
| 1989 | 8 | 30 | 1.16 |
| 1990 | 8.3 | 31 | No data |
| 1991 | 8.3 | 15 | 0.191 |
| 1992 | 8.2 | 23 | 0.191 |
| 1993 | 8 | 90 | No data |
| 1994 | 8 | 63 | 0.328 |
| 1995 | 7.9 | 89 | No data |
| 1996 | No data | 88 | No data |
| 1997 | 8 | 89 | No data |

**Figure 5.** Average values of water quality parameters of the studied surface water body and chemical conditions typology as stated by ECOFRAME and MED-GIG system (data were provided by HMRDF 2019, [a] Criteria given by ECOFRAME [15]; [b] Criteria given by Poikane [16]).

## 4. Conclusions

The studied water body was subjected to particularly high pressures, mainly due to agricultural activities. The data analysis of physical and chemical elements indicated that a high concentration of nutrients occurs in the water body, maintaining the eutrophic conditions of Lake Vegoritida. This study found that Lake Vegoritida is classified as "Good to Moderate" and tends to change to the "Moderate" class, suggesting the need for restoration and prevention projects supported by the installation of the real-time monitoring network of water quality. The combination of ANN and DPSIR approach proved to be an efficient and helpful tool for evaluating the chemical status of a surface water body. Moreover, the combination of the DPSIR model and ANN applications may be helpful for stakeholders and policymakers for applying a sustainable development management of surface water resources.

**Author Contributions:** Conceptualization, D.E.A., C.T. and K.M.; Methodology, D.E.A., C.T. and K.M.; Software, C.T.; Validation, K.M. and D.E.A.; Formal Analysis, C.T., D.E.G. and K.M.; Investigation, C.T., D.E.A. and D.E.G.; Resources, D.E.A. and D.E.G.; Data Curation, C.T., K.M. and D.E.A.; Writing—Original Draft Preparation, C.T., D.E.A., K.M. and K.M.; Writing—Review and Editing, D.E.A. and K.M.; Visualization, C.T.; Supervision, D.E.A. and K.M.; Project Administration, D.E.A. and K.M. All authors have read and agreed to the published version of the manuscript.

**Funding:** This research received no external funding.

**Data Availability Statement:** The data used in this study are available to external parties upon request at the Greek Ministry of Rural Development and Food: http://www.minagric.gr/ (accessed on 7 May 2022).

**Acknowledgments:** The authors would like to thank the Greek Ministry of Rural Development and Food for kindly providing valuable data for this research.

**Conflicts of Interest:** The authors declare no conflict of interest.

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
