# Peer review of "Combining Artificial Neural Network and Driver–Pressure–State–Impact–Response Approach for Evaluating a Mediterranean Lake"

_water, doi:10.3390/w15020266_

Round 1

Reviewer 1 Report

The paper presents the evaluation of the surface water system of a lake using an artificial neural network for the prediction of the water quality. This work is very interesting in terms of an overall ecological assessment. The experimental results verify that the artificial neural network is a powerful tool for the assessment of the water quality. The method as well as the experimental results are well presented and discussed. I only suggest that the authors should include a diagram concerning the convergence of the artificial neural network to highlight its validity. In addition, the text should be checked again considering some typos: pg.1, ln.36, pg.2, ln.59, pg.5, ln.200,204, pg.6, ln.221.

Author Response

Response to Reviewer 1 Comments

Point 1:. I only suggest that the authors should include a diagram concerning the convergence of the artificial neural network to highlight its validity. In addition, the text should be checked again considering some typos: pg.1, ln.36, pg.2, ln.59, pg.5, ln.200,204, pg.6, ln.221.

Response 1: Thanks for your comments, we include a diagram concerning the convergence of the artificial neural network. The typographical errors noted in the text are corrected.  

Τhe value of determination coefficient (R2) is equal to 0.962, which shows that the developed ΑΝΝ model has the ability to interpret 96.2% of the variation in the average annual (TP) concentration values for Lake Vegoritida. The value of agreement index (IA) is equal to 0.986, very close to unity, indicating that the predicted values of the average annual (TP) concentration from the model are very close to the corresponding observed values.

Figure 3. Determination coefficient beteewen observed and predicted TP values by ANN.

Reviewer 2 Report

The authors have contributed to science by producing a paper documenting their analysis of temporal water quality data collected from a lentic water body. They have evaluated the usefulness of various schemes used to identify the ecological/biological condition of lakes based on water quality data. Their findings will be useful for successful future management of the catchment area of lakes to reduce the negative impact of anthropogenic activities such as fertilizer usage.

However, the manuscript isn’t publishable in its current form because the English requires further editing to improve clarity, spelling, and grammatical correctness. I haven’t made specific comments past the first lines of the abstract.

Manuscript ID: Water-2094342

Review comments/suggestions

Be consistent with the name of the lake is it Lake Vegoritida (in the Abstract) or Vegoritis Lake?

I suggest minimizing the use of abbreviations (where possible) to improve readability of the text and interpretation of tables and figures.

For consistency it would be a good idea to use either “mean” or “average” rather than a mixture of terms.

The units water quality parameters were measured in have been omitted from the Methodology section.

Abstract

Page 1, Lines 12 - 17

The main objective of this research is was the evaluation of the surface water system of Lake Vegoritida, (Region of Central Macedonia, Greece). Selection of using the Driver-Pressure-State-Impact-Response (DPSIR) methodological approach. The analysis includes Data collected from (3) stations monitoring point source pollution and by recording the most important water quality measurement parameters in a time series analysis of data from 1983 to 1997. The data will contribute to the analysis and was used to investigation of identify and evaluate possible sources of chemical and ecological changes of recorded in the lake. The Artificial Neural Network (ANN) proved to be a valuable tool for predicting  making predictions based on the water quality data set.

Figure 1. If available it would be helpful if the location of the water chemistry monitoring stations on the lake were indicated on the map. For readers who are unfamiliar with Lake Vegoritis, an inset with a map of Greece indicating where this lake is would help set the scene for your study.

Table 1. would benefit from the text being aligned left rather than centred. I suggest replacing “lowest” with “Minimum” or use “Highest” and “Lowest”

Could Tables 2. and 3. be included in Supplementary data and summary tables be used to highlight key findings?

The number of significant figures varies in Table 2., as does the format e.g. TP     p-value  .0071     0,157     0,106

Table 5. would benefit from the text being aligned left rather than centred.

Figure 4. looks more like a Table than a Figure.

The references listed aren’t in a consistent format.

Author Response

Response to Reviewer 2 Comments

Point 1:. Be consistent with the name of the lake is it Lake Vegoritida (in the Abstract) or Vegoritis Lake?

Response 1: Thank you for this comment. We have corrected this issue.

Point 2: I suggest minimizing the use of abbreviations (where possible) to improve readability of the text and interpretation of tables and figures.

Response 2: The comment for the Referee is fair. The use of abbreviations is minimised to improve readability of the manuscript.

Point 3: For consistency it would be a good idea to use either “mean” or “average” rather than a mixture of terms.

Response 3: Thanks for your comment. Only the term "average" appears in the text now.

Point 4: The units water quality parameters were measured in have been omitted from the Methodology section.

Response 4: The comment for the Referee is fair. The units for each quallity parameter are now included in the Methodology section.

The text is revised and now reads:

“The following water quality parameters: Total Nitrogen (TN) mgL-1, Total Phosphorous (TP) μgL-1,dissolved oxygen (DO) mgL-1, pH and Electrical Conductivity (CND) μs cm-1.”

Point 5: Page 1, Lines 12 - 17

The main objective of this research is was the evaluation of the surface water system of Lake Vegoritida, (Region of Central Macedonia, Greece). Selection of using the Driver-Pressure-State-Impact-Response (DPSIR) methodological approach. The analysis includes Data collected from (3) stations monitoring point source pollution and by recording the most important water quality measurement parameters in a time series analysis of data from 1983 to 1997. The data will contribute to the analysis and was used to investigation of identify and evaluate possible sources of chemical and ecological changes of recorded in the lake. The Artificial Neural Network (ANN) proved to be a valuable tool for predicting making predictions based on the water quality data set.

Response 5: Thank you very much for your help. Your suggestions help us to enhance the quality of abstract.

The abstract is revised and now reads:

“Abstract: The main objective of this research was the evaluation of the surface water system of Vegoritida Lake, (Region of Central Macedonia, Greece). Selection of using the Driver-Pressure-State-Impact-Response (DPSIR) methodological approach. The analysis includes data collected from (3) stations monitoring point source pollution by recording the most important water quality measurement parameters in a time series analysis of data from 1983 to 1997. The data will contribute to the analysis and was used to investigation of identify and evaluate possible sources of chemical and ecological changes of recorded in the lake. The artificial neural network (ANN) proved to be a valuable tool for making predictions based on the water quality data set”

Point 6: If available it would be helpful if the location of the water chemistry monitoring stations on the lake were indicated on the map. For readers who are unfamiliar with Lake Vegoritis, an inset with a map of Greece indicating where this lake is would help set the scene for your study.

Response 6: Thanks for the comments, we include a map of Greece (a) and the locations of monitoring stations for Lake Vegoritida (b)

Figure 1. (a) Map of Greece showing the location of Lake Vegoritida, (b) Map of study area showing the location of monitoring stations.

Point 7: would benefit from the text being aligned left rather than centred. I suggest replacing “lowest” with “Minimum” or use “Highest” and “Lowest”

Response 7: It is revised. Please see the manuscript.

Point 8: Could Tables 2. and 3. be included in Supplementary data and summary tables be used to highlight key findings?

Response 8: Thanks for this comment. Unfortunately, Tables 2 and 3 cannot be merged because they include corelation results derived form different monitoring periods.

Point 9: The number of significant figures varies in Table 2., as does the format e.g. TP     p-value  .0071     0,157     0,106

Response 9: All the typographical errors are now corrected.

Point 10: Table 5. would benefit from the text being aligned left rather than centred.

Response 10: The comment is fair. The text is aligned.

Point 11: Figure 4. looks more like a Table than a Figure.

Response 11: This comment needs further discussion. We selected this format to follow the methodology suggested by WFD to classify the status of a surface water body using coloured blocks. This approach is not compatible with the Table format suggested by MDPI. Concerning the authors template we selected to include all the appropriate information in a figure format because we can use colours to highlight the status of the water body. However, if the Referee insists on his initial comment, we will treat this issue accordingly.

Point 12: The references listed aren’t in a consistent format.

Response 12: Thank you it is revised.

Reviewer 3 Report

Water-209432

Title:Combining Artificial Neural Network and Driver-Pressure-State-Impact-Response Approach for Evaluating a Mediteranean Lake

Authors: Christos Tsitsis1 , Dimitrios E. Alexakis, Konstantinos Moustris, Dimitra E. Gamvoula

The study is to evaluation of the surface water system of Lake Vegoritida in Greece with the Driver-Pressure-State-Impact-Response (DPSIR) approach.

Main comments:

The paper was not well organized and written, and no innovation methodology is developed. The data analysis, and DPSIR approach are results from the study, which more like a report, not a technical paper.

Abstract: There are so many grammar issues and very hard to understand. It would benefit gratly from copy-editing. For example:

Line 13-14, is not a sentence

Line 17,  proved--à is proved

Line 20-22, not a sentence

Line 22 -24, two sentences

Line 26 -28, repeated Line 29-31.

Introduction:  The first paragraph should move to the end after the problem statement are clearly presented.

Line92-93, has been repeated at Line36-37

Methods:

An ANN is used to fit the missing values of the recorded data. It is a widely used tool and no need to have Figure 2 copied from the [33]. However, more detailed explaination is expected on what data was trained, how the model was developed, what accuracy was reached?

Results:

To me, 3.1 is the literature review, should be summaried in “introduction”.

3.4 should be part of the “Study area"

Conclusion:  This is more like a summary, not a conclusion.

Author Response

Response to Reviewer 3 Comments

Point 1:. The paper was not well organized and written, and no innovation methodology is developed. The data analysis, and DPSIR approach are results from the study, which more like a report, not a technical paper.

Response 1: The authors would like to sincerely thank Reviewer 3 for carefully reviewing our manuscript and providing us with comments and suggestion to enhance the quality of the manuscript. The following responses have been prepared in order to address all of the reviewers’ comments, point-by-point. While revising the manuscript in accordance to the reviewers’ suggestions, the manuscript was also revised regarding some minor grammatical and syntax details.

 The findings of this study will contribute to the international database of investigations on surface water quality assessment based on the DPSIR model and ANN applications. The novelty of this research study lies in the use of a combination of DPSIR model and ANN applications for sustainable development management of surface water resources. The findings of this research may be helpful for stakeholders and policymakers monitoring agricultural areas. No other previous studies report results of the DPSIR model or ANN approach in the Vegoritis lake to the best of the author knowledge.

Point 2: Abstract: There are so many grammar issues and very hard to understand. It would benefit gratly from copy-editing. For example:

Line 13-14, is not a sentence

Line 17,  proved--à is proved

Line 20-22, not a sentence

Line 22 -24, two sentences

Line 26 -28, repeated Line 29-31

Response 2: Thanks for your help. All the comments are adoptied. Please see the revised version of the abstract.

Point 3:. Introduction: The first paragraph should move to the end after the problem statement are clearly presented.

Response 3: Thank you for your suggestion the Introduction is revised.

Point 4: Line92-93, has been repeated at Line36-37

Response 4: It is revised

Point 5: Methods:

An ANN is used to fit the missing values of the recorded data. It is a widely used tool and no need to have Figure 2 copied from the [33]. However, more detailed explaination is expected on what data was trained, how the model was developed, what accuracy was reached?

Response 5: We have included Figure 3 which shows the prediction accuracy of ANN. Additionally, Figure 3 from Moustris et. al. [33] was reported because the applied methodology in this study uses the same trainning model as shown in Figure 2.

“The final architecture of the developed ANN model consists of an input layer with three (3) input artificial neurons, a hidden layer with three (3) hidden (ANN), and an output layer with one (ANN) corresponding to the rate of mean annual (TP) concentration. The selection of the appropriate input data as well as the architecture of the developed ANN was found after iterative application of the trial and error method [34,35]. Τhe value of determination coefficient (R2) is equal to 0.962, which shows that the developed ΑΝΝ model has the ability to interpret 96.2% of the variation in the average annual (TP) concentration values for Lake Vegoritida. The value of agreement index (IA) is equal to 0.986, very close to unity, indicating that the predicted values of the average annual (TP) concentration from the model are very close to the corresponding observed values”.

Figure 3. Determination coefficient between observed (TP) values and predicted (TP) values by ANN.

Point 6:. Results: To me, 3.1 is the literature review, should be summaried in “introduction”.

Response 6: Thank you for the suggestion the 3.1 section moves now to the Methodology section.

Point 7: 3.4 should be part of the “Study area"

Response 7: Thank you for your suggestion the title of the sub-section is revised and now reads:

“3.3. Typification of Lake Vegoritida”. This sub – section includes the results derived by the classification on the studied water body according to the Ecoframe approach.

Point 8: Conclusion: This is more like a summary, not a conclusion.

Response 8: Thank you for your suggestion. The Conclusions section is revised to fulfill the Reviewer’s comment.

Round 2

Reviewer 2 Report

The manuscript (ID: Water-2094342 V2) has undergone a revision and significant improvement is apparent. All comments have been addressed. Your revised version has been reviewed and suggested minor changes are outlined below.

Review comments/suggestions

Abstract

Page 1, Lines 13 - 14

The Driver-Pressure-State-Impact-Response (DPSIR) methodological approach was used.

Page 1, Lines 16 - 17

The data will contribute to the analysis and was used to investigation of identify and evaluate possible sources of chemical and ecological changes of recorded in the lake.

Page 1, Lines 26 - 27

The Actions is required at an early-stage concern in the planning of programs and actions that contributing to the appropriate management of land uses, …………

Introduction

Page 1, Line 32

leaching, which cause adverse environmental relegation degradation……..

Page 1, Line 40

Communities and their activities increase rapidly increase the pressures exerted on……..

Page 1, Lines 42-44

….ecological elements of the region to provide ensure sustainability, the quality of water resources, and the natural environment, as well as and a more rational balanced coexistence of the natural component environment in combination with human activities.

Page 2, Line 46

……..adoption of the Water Framework Directive (WFD) by the European……..

Your readers may not be in Europe and be unfamiliar with the abbreviation WFD. Include the full title once with the abbreviation in brackets then just WFD in the remainder of your paper.

Page 2, Lines 47- 48

WFD is was to create the…….. state by 2015.

If 2015 is the correct date the objective set by the WFD is now in the past.

Page 2, Lines 51-54

Currently, Many surface fresh waters bodies worldwide face strong degradation dramatic environmental degradation with downward trends of in water quality degradation, and various studies support predicted that many lakes in the European region will would encounter difficulties in implementing the required WFD criteria until by 2015.

This information is now 7 years old, so I suggest remove the word “Currently” and write in the past tense.

Page 2, Line 57

……temperate limno lentic systems, considering lake systems, studying….

Page 2, Line 63

 ……..a itemized standardized framework for identifying……..

Page 2, Line 73

The investigated water body (Lake Vegoritida)……….

Page 2, Lines 76 - 77

……the lake presents an unfavorable quality of its has an ecosystem of poor biological quality while at the same time, ……… recreation) are still permitted to take place…

Page 2, Lines 79-80

the involvement to involve the formulation of contemporary management……..

Methodology

Page 2, Line 87

The lake captures an has a surface area of about approximately ……….

Page 3, Line 98

situation condition of the lakes has become  became a legal obligation after the according introduction of the WFD.

Page 3, Line 106

Total Phosphorous (TP) μmg L-1

Page 3, Line 107

………(CND) μS cm-1

Page 3, Line 113

biological characteristics indicators (bioindicators) since plant, invertebrate and fish species interact and affect……

Page 3, Line 116

…..different biological teams groups……

Page 3, Lines 116-117

procure the possibility provide an opportunity to assess……

Page 4, Lines 127-128

Some countries in their area include many small or large lakes while other countries of the same region have only small or large lakes.

I suggest the following:

The number and size of lakes within countries located in the European region varies greatly.

Page 4, Line 129

…..mainly used for water storage and the supply of water for agricultural purposes.  work.

Page 4, Lines 132-132

……change, whereas is in the northernmost part of the continent, we find areas with less population populated with and less lower pressure…

Page 4, Line 141

…(b) the fluctuation of the water level as a cause result of natural……

Page 4, Line 149

….also common. practices

Page 4, Line 151

Early in the 1950s, the development of artificial neural networks (ANNs) as…..

Page 5, Figure 2 caption Line 168

Figure 2. Typical architecture of a feed-forward Multi-Layer-Perceptron Artificial Neural Networks (MLP – ANN).

It’s good practice to have self-explanatory captions so your readers find interpretation easier.

Page 6, Line 208

…..are nether lower than the average DO concentrations…..

Page 6, Line 212

…..Lake Vegoritida Vegoritis lake, since according to [18] and [3].

Page 7, Line 225

…….average TP values are presented were recorded …….

Page 7,  Table 1.

Conductivity  S cm-1)   TP ( μmg L-1)

Page 10, Lines 261- 262

Reference conditions are defined based on which any deviation of the ecological status…..

……water quality that is related to the change of conditions……

Pages 10-11, Table 4.

Vegoritis Lake Lake Vegoritida

Hightest and average depth

The lake possess  has an………

Page 12, Figure 5. Column 3 label

TP ( μmg L-1)

Page 13, Line 312

It is worth noting that at in the Amyntaio and Ptolemaida area 270 and………

Page 13, Line 334

The data analysis through the of physical and chemical elements…….

Author Response

Response to Reviewer 2 Comments

Point 1: The manuscript (ID: Water-2094342 V2) has undergone a revision and significant improvement is apparent. All comments have been addressed. Your revised version has been reviewed and suggested minor changes are outlined below.

Response 1: Thanks for your suggestions and constructive comments. Αll the suggestions are adopted. Your comments helped us to further enhance the quality of the manuscript.
Regards,
